# Multi-party co-signature scheme based on SM2

Liang Tan[1¤c], Xinglin Shang[1], Liping Zou[1☺¤a], Hekun Yang[1☺¤b], Yi Wen[1], Zhongzhu Liu[2]*

1 College of Computer Science, Sichuan Normal University, Chengdu, Sichuan, China, 2 School of Mathematics and Statistics, Huizhou University, Guangdong Province, China

☺ These authors contributed equally to this work.
¤a Current address: Southwest Petroleum University, Network and Information Center, Chengdu, Sichuan, China
¤b Current address: Department of Computer Science, Sichuan Aerospace Vocational College, Chengdu, Sichuan, China
¤c Current address: Institute of Computing Technology, Chinese Academy of Sciences, Beijing, China
* zhongzhuliu@126.com

**Data Availability Statement:** The data underlying the results presented in the study are available from the Open Science Framework database (https://osf.io/BHX32/).

**Funding:** This work was supported by the National Natural Science Foundation of China 61373162, the

## Abstract

Two-party collaborative signature scheme is an important cryptographic technology for user authentication and data integrity protection when using mobile devices for financial and securities transactions. However, the two-party collaboration scheme has the following shortcomings: firstly, it is not flexible enough, and it requires the collaborating parties to be secure and trusted; secondly, the two-party collaboration security still needs to be improved. Once a hacker obtains the signature private key and collaborative identity of a mobile device, it can construct a legitimate two-party collaborative signature. Third, the application scenario of two-party co-signature is limited and cannot meet the application scenario of multi-device co-signature. For this reason, this paper designs a multi-party collaborative signature scheme based on SM2 digital signature algorithm in the standard "SM2 Elliptic Curve Public Key Cryptography" of GM/T003-2012. This scheme consists of multiple (more than 2) participants to jointly generate the signature group public key and valid signature in an interactive manner, while ensuring that each user cannot know the signature key other than their own during the signing process. We implement this scheme based on the GMP library. The experimental results show that this scheme is not only flexible but also more secure and trustworthy to meet the application scenario of multi-device collaborative signing. In addition, the time for multiple participants to construct signatures in this scheme is similar, and the time for signature verification is less different from that of the original SM2 signature.

## Introduction

With the rapid development of mobile Internet, the owners of mobile smart devices are increasing, and device owners can use their mobile devices to communicate with each other anytime and anywhere, and mobile devices have been fully integrated into the production life

Sichuan Provincial Science and Technology Support Project 2019YFG0183, and the Sichuan Provincial Key Laboratory Project KJ201402. The funder had no role in study design, data collection and analysis, decision to publish, or preparation of the manuscript.

**Competing interests:** The authors have declared that no competing interests exist.

of society. According to real-time data from GSMA intelligence, as of the end of 2019, there are currently more than 5.2 billion unique owners of mobile devices worldwide (i.e., more than 67% of the world's population has a mobile device), and this number is forecast to increase to 5.8 billion by the end of 2025, accounting for 70% of the world's population [1]. Nowadays, it is very common for users to use mobile devices to conduct financial, securities and other transactions, and it has become particularly important to ensure the security of sensitive user data and transaction processes [2–4].

Data signature is an important cryptographic technology for authentication of mobile device users and integrity protection of mobile device data, which can play an important role in protecting users' sensitive data and transaction process security. However, the digital signature technology has extremely high requirements for the storage and management of the signature private key of the mobile device, mobile devices themselves have security risks such as easy loss or hijacking, and limited computing power, if the software module is used to save the signature private key to the local or smart chip [5], this simple device security deployment and open network connections make mobile devices extremely easy to become the target of network attacks [6, 7]. Once the mobile device is hacked and the signature private key is stolen, the hacker can pretend to be a user in banking, insurance, securities, transportation, postal, e-commerce, mobile communications and other industries to conduct transactions, causing huge economic losses to the user.

Currently, there are two specific methods to solve this problem. One is the threshold signature scheme based on Shamir's secret sharing, which splits the signature private key of a device among $n$ participants when and only when there are more than a threshold $t$ ($t \leq n$) participants collaboratively recovering the complete private key and signing it with their respective private key slice [8]. The signature private key slice in this scheme is no longer stored and managed by a unique device but by n devices, thus it is more secure and trustworthy, and even if a hacker breaks one or $m (m < t)$ of the devices and obtains their private key slice, it still cannot construct a legitimate signature. However, this scheme is difficult to apply to mobile terminals with limited computing and storage resources. In addition such threshold signature schemes [9–13] all require the participation of trusted centers and suffer from high communication computational cost, complex interaction, and too many communications. The second is the two-party co-signature scheme. The so-called two-party co-signature scheme refers to the co-signature scheme in which only 2 entities participate, usually one is a mobile terminal, and the other is a secure and trusted server. There are also some research results available. The two-party ECDSA signature scheme proposed by Ref. [14] in 2017 provides a good research direction for the subsequent work [14–16]. The literature [17–19] proposed a two-party co-signature protocol based on the SM2 algorithm. The Ref. [20] proposed a two-party co-signature scheme based on the SM2 algorithm, which gives a security model and a proof of security, with only one communication between the signing parties. The signature generated by the two-party co-signature scheme is no longer generated by the mobile terminal device alone, but is generated by the two parties together, which greatly increases the difficulty for hackers to construct a legal signature and improves the security and trustworthiness of the signature.

However, the two-party co-signing scheme still has three problems: first, two-party co-signing is not flexible enough. It requires that the cooperating party must be secure and trustworthy, so that the constructed collaborative signature can ensure the identity authentication and data integrity of mobile users. second, the two-party co-signing security still needs to be improved. Once a hacker obtains the signature private key and collaborative identity of the mobile device as the initiator of the signature, it is able to construct a legitimate two-party collaborative signature. Third, the application scenarios of two-party collaborative signatures are limited, which cannot meet the application scenarios of multi-device collaborative signatures.

For example, in e-government affairs, leaders at all levels need to approve and sign the same document from lower to higher levels. In e-commerce Multiple partners sign the same contract, and the order of signatures can be fixed or random, etc.

To this end, this paper proposes a scalable multi-party collaborative signature scheme based on the SM2 algorithm. In this scheme, each participant generates a key pair, and any participant can initiate a signature, and multiple parties jointly participate in generating a valid signature and a group public key for signature verification, which can ensure that each participant cannot know the signature key other than their own during the signing process, and the original complete signature cannot be recovered by any missing user. The experimental results show that this scheme can not only construct a legitimate signature, but also the time for multiple participants to construct a signature is similar, and the signature verification can pass and the verification time is less than the original SM2 scheme. Moreover, it is more flexible, safer and more reliable, and is suitable for application scenarios of multi-device collaborative signature, which has great practical value in events that require the joint participation of multiple departments, such as document issuance, certificate issuance and auditing.

This article is divided into 7 sections for introduction. Section 1 introduces the background and development of SM2 algorithm, Section 2 introduces the basics involved in SM2 algorithm, Section 3 introduces the model and objective of SM2 algorithm, Section 4 introduces the principle of SM2 multi-party collaborative signature scheme based on SM2, Section 5 analyzes the security of SM2 multi-party collaborative scheme, Section 6 introduces the experiments and results analysis of SM2 multi-party collaborative scheme. Section 7 concludes the scheme of this paper.

## Basic knowledge

This section will introduce the basic knowledge of digital signature and collaborative computing involved in this article. The symbol descriptions of related parameters are as follows.

In this paper, $\lambda$ denotes the safety parameter and $\mu(\lambda)$ denotes a negligible function of $\lambda$. In addition, [*] denotes the dot product operation of elliptic curve and [−] denotes the dot subtraction operation of elliptic curve; $\odot$ denotes the scalar multiplication homomorphism operation, that is, $a \odot b$ denotes the plaintext corresponding to $b$ does multiplication operation with $a$; $\oplus$ means addition homomorphic operation, that is, $a \oplus b$ means that the plaintext corresponding to $a$ and the plaintext corresponding to $b$ are added. $H_v()$ is the hash algorithm with digest length $v$ bits. $Encpk()$ denotes the encryption operation of the Paillier scheme [21] under the homomorphic public key $pk$, and $Decsk()$ denotes the decryption operation of the Paillier scheme [21] under the homomorphic private key $sk$.

**Definition 1**. $F_p$ is a finite field, $p$ denotes the scale of the finite field, and p is an odd prime or a square power of 2. E denotes an elliptic curve over a finite field $F_p$, and choose $a, b \in F_p$ as the parameters of the elliptic curve $E$, where the elliptic curve $E(F_p)$ satisfies $y^2 = x^3 + ax + b \bmod p$ and $(4a^3 + 27b^2) \bmod p \neq 0$, and $G$ is a point on the elliptic curve, that is, $E(F_p) = \{G | G \in E\} \cup \{O\}$. A set formed by the addition operation of $G$ as the generator point is called a group [22], the number of points in this group is called order $n$, where $\{O\}$ is the point at infinity.

## SM2 digital signature

According to the specification of "SM2 Elliptic Curve Public Key Cryptography Algorithm", the definition of SM2 digital signature [23] is as follows:

(1) **Parameter initialization (Setup)**: Input security parameters $k$, generate elliptic curve parameters $params = (p, a, b, G, n)$ and output.

(2) **Key generation (Key)**: Given the elliptic curve parameters *params*, select the random number $d_A$ as the user's private key and calculate $P = d_A \times G$ as the corresponding public key. Output the public-private key pair $(P, d_A)$.

(3) **Signature algorithm (Sign)**: Given the elliptic curve parameters *params*, the private key $d_A$ and the message *m*, the algorithm steps for generating the signature $(r, s)$ are as follows:

① Calculate $Z_A = H_v(ENTL\|ID_A\|a\|b\|P\|pk)$, where $ID_A$ is the distinguishable identifier of the user, *ENTL* is the length of $ID_A$, calculate $M' = ZA\|M$;

② Calculate $e = H_v(M')$, and convert the text *e* to an integer, where $H_v()$ is a one-way function;

③ Select the random number $k \in Z_p^*$, calculate the elliptic curve point $Q = k[^*]G = (x_1, y_1)$, and convert the data type of $x_1$ to an integer;

④ Calculate the signature $r = (e + x_1) mod\ n$, if $r = 0$ or $r + k = n$, return ③;

⑤ Calculate the signature $s = (1 + d_A)^{-1}(k - rd_A) mod n$, if $s = 0$, return to ③; otherwise, output the signature result $(r, s)$.

(4) **Signature verification (Verify)**: Given parameters *params*, public key $P = d_A[^*]G$, message $m'$ and signature $(r', s')$, the steps of the verification algorithm are as follows.

① Respectively check whether $r', s' \in [1, n - 1]$ is established, if not, the verification fails;

② Calculate $M'' = Z\|M'$, compute $e' = H_v(M')$ and convert the data type of $e'$ to an integer;

③ Convert the data type of signature $(r', s')$ to integer and calculate $t = (r' + s')\ mod\ n$. If $t = 0$ then the verification does not pass; otherwise, go to step ④;

④ Calculate the elliptic curve point $(x_1', y_1') = s'[*]G + t[*]P$ and convert the data type of $x_1'$ to an integer;

⑤ Calculate $R = (e' + x_1')\ mod\ n$, check whether $R = r'$ is established, and output 1 if the verification passes; otherwise output 0.

## Secure Multi-Party Computation

Secure Multi-Party Computation (SMC) was first proposed by Prof. Yao in 1980 [24], which is mainly used to solve the personal privacy problem between participants who do not trust each other.

Secure Multi-Party Computation means that neither a public random string nor complex distributed settings are required. Each participant can generate its own key pair completely independently without any auxiliary information to complete the computation, which is an effective method to solve the privacy computation problem between two or more participants with strong theoretical and practical significance. In this paper, the scheme consists of multiple participants each selecting their own key pairs and jointly completing the signatures without revealing their respective privacy information, and during the computation process, all participants cannot obtain any additional valid information except for knowing their respective input data (including the received data transmitted by others) and the returned data results.

## Safety model and design goals

Based on the co-signature described in the introduction, we construct the following application scenarios. Taking a customer's loan application as an example, a bank has three-level administrative departments A, B, and C. When a customer makes a large loan in the bank, the approval process will be relatively strict, and two or more departments are often required to review the transaction documents level by level [25], and the specific steps for multi-level signing of transaction documents by multiple departments are as follows.

1) The customer initiates an application for a large loan with the bank.

2) First, the bank receives the application, and the third-level department C reviews the customer's personal credit and other information, and if the basic information is legal, the transaction document will be signed for the first time.

3) Then, after receiving the transaction documents signed by the tertiary department, the secondary department B will conduct a valuation review on the client's asset information, etc. If the review is approved, the transaction documents will be signed for the second time.

4) Finally, the Level 1 department summarizes and finally reviews all the information required by the customer, and if there is no problem, the transaction document will be signed for the third time.

In the above application, which involves joint cooperation among 3 departments, more review steps and corresponding signatures may be required in practical application scenarios. Therefore, this section will detail the security model on which the digital signature is based and the ideal design goals of the scheme in this paper. The main goal of the digital signature algorithm security model is to ensure that the attacker cannot obtain the user's private data even through multiple signature interactions.

## Safety model

The scheme in this paper is based on the framework of the MPC security proof model. By constructing an analog protocol, using the interaction of the signature process, the security is regulated to the security proof of the original digital signature scheme [24, 26].

**(1) Communication model.** It is assumed that the system parameters (and the standard parameters of SM2) can be passed to all participants in an efficient and secure manner. In addition, there is a peer-to-peer channel connected to any 2 participants to carry the protocol interactions.

**(2) Multiple digital signature.** Depending on the signature process, multiple digital signatures [27, 28] are classified into sequential and broadcast multiple signatures. Sequential multi-signature means that the message sender specifies the order of signing the message, and then sends the message to be signed to the first signing member. After each signing member receives the message, it first verifies the validity of the previous signing member's partial signature on the message, and if it is valid, it continues to sign and sends the generated partial signature to the next signing member; if the signature is invalid, it refuses to continue signing the received message and terminates the whole signing process. Until the last signing member completes the partial signature of the message, that is, the sequential multi-signature is completed [29]. In this paper, the key generation and co-signing parts of the scheme involve chain calculation data, so the concept of multiple digital signatures is used to complete the chain calculation of data.

**(3) Safety model.** The security model of the digital signature algorithm contains challenger and adversary. The main attack on the digital signature scheme is forging the signature. If adversary is unable to forge a digital signature after several attacks, the signature constructed by challenger is considered to be secure.

**Definition 2**. A signature algorithm (*Setup, Key, Sign, Verify*) is given in the digital signature algorithm attack game. Challenger runs the *Setup* and *Key* algorithms and exposes the generated system parameters *params* and the verifying public key *P*. Adversary obtains the parameters and initiates $n$ ($n$ is large enough) signature training to challenger. That is, the input message sequence $\{M_1, M_2, \ldots, M_n\}$ requires challenger to give a legitimate digital signature $\{\delta_1, \delta_2, \ldots, \delta_n\}$. After $n$ times of signature training, adversary is able to construct a valid signature message pair $(M^*, \delta^*)$ satisfying the condition $M^* \neq \{M_1, M_2, \ldots, M_n\}$ and *Verify* $(M^*, \delta^*) = 1$. It is considered that the adversary successfully forged the signature, that is, the

signature constructed by the challenger is not secure, and the event is called *Event_attack* [2] and the *Event_attack* event satisfies the Eq (1).

$$Event\_attack\{$$

$$S : \{M_1, M_2, ..., M_n\} \rightarrow \{\delta_1, \delta_2, ..., \delta_n\} \ \&\& \ Verify(M_i, \delta_i) = 1$$

$$A : \{M^*\} \rightarrow \{\delta^*\} \ \&\& \ Verify(M^*, \delta^*) = 1$$

$$\}$$

$$(1)$$

Conversely, if all signature message pairs $(M^*, \delta^*)$ constructed by adversary cannot satisfy the condition $M^* \notin \{M_1, M_2, \cdots, M_n\}$ and $Verify(M^*, \delta^*) = 1$, then adversary is considered unable to forge a signature and this event is said to be a safety model of *Event_safe*, and the *Event_safe* event satisfies Eq (2).

$$Event\_safe\{$$

$$S : \{M_1, M_2, ..., M_n\} \rightarrow \{\delta_1, \delta_2, ..., \delta_n\} \ \&\& \ Verify(M_i, \delta_i) = 1$$

$$A : \forall \, M^* \notin \{M_1, M_2, ..., M_n\}, \{M^*\} \rightarrow \{\delta^*\} \&\& \ \exists \ Verify(M^*, \delta^*) \neq 1$$

$$\}$$

$$(2)$$

## Design goals

The existing scheme enables multiple users or multiple devices $U_1, U_2, ..., U_m$ to jointly complete a digital signature and satisfy a security model, where $m>2$ and any legal member of the signature can be signature initiator. In this model, participants in collaborative signatures must meet the following conditions.

(1) The public keys of all participant key pairs are made public before participating in co-signing, and each participant has a unique index number for identity identification.

(2) The public key values of all participants should be different, and even if the attacker and the participants have the same public key, they still correspond to multiple different private key values, which is guaranteed by the difficulty of solving the discrete logarithm problem on the elliptic curve.

(3) Two steps are required to process the transmitted data during chain computing: ① encrypting the transmitted data with the next user's public key value; ② digitally signing the encrypted data to be transmitted with the current user's private key.

(4) As long as the participant is a legal member of the collaborative signature and participates in the chain calculation of the public key value of the group, it is necessary to unconditionally cooperate with the signature when initiating the signature;

(5) The attacker can corrupt up to m-1 participants, that is, at least one participant has not been corrupted, this scheme proves to be meaningful.

The condition (1) is to distinguish whether it is a legal member of the collaborative signature, and the condition (2) can ensure that the encrypted data can only be decrypted by the only user who knows the corresponding private key, so as to ensure the confidentiality of the scheme; the private key in condition (3) decrypts the cryptographic data to guarantee the non-forgeability of the transmitted data, as well as the non-repudiation of the operation guaranteed by the digital signature verification, and from the multiple digital signatures in Section 3.1, it is known that all participants have and only one participation in the chain calculation, condition

(4) can guarantee the correctness of the digital signature, and condition (5) can guarantee the integrity and practical significance of the multi-party co-signature scheme.

## SM2-based multi-party co-signing scheme

In the design scheme of this paper, the signature is completed jointly by $m$ participants $U_1$, $U_2$, . . ., $U_m$. This section will introduce in detail the SM2-based multi-party collaborative signature scheme and the specific algorithm implementation proposed in this paper. The multi-party co-signing scheme is divided into two parts: key generation and co-signing, in which the generation of the group public key $P$ in the key generation protocol and the signature intermediate value $D$ in the co-signing process need to use the idea of sequential multi-signing in Section 31 to complete the data transmission and computation, and the sequential computation here is only to ensure that all co-signing participants participate in the computation only once. In the multi-party co-signing scheme, the system parameters and elliptic curves of the SM2 signature scheme are not changed. For the system initialization part of SM2, refer to the specification of "SM2 Elliptic Curve Public Key Cryptography Algorithm".

### Scheme design

In this scheme, the parameter initialization and signature verification process are the same as the national standard SM2 algorithm, which makes the scheme more applicable and has more application scenarios. The multi-party collaborative signature scheme is jointly completed by $m$ ($m \geq 3$) participants. First, each of the $m$ participants invokes a function to generate a key pair, and the $m$ participants chain invoke the function to compute the public key $P$ of the signature group. Then the participants jointly sign, and any participant can become the signature initiator, and the final signature result ($r$, $s$) is calculated by the signature initiator. The symbol descriptions of related parameters are as follows: elliptic curve parameters are represented as $params = (p, a, b, G, n)$; $U_i$ represents the $i$-th participant, $d_i$ represents the private key of the participant $U_i$, and $P_i$ denotes the public key of participant $U_i$, $k_i$ represents the random number of participant $U_i$, $K_i$ represents the intermediate value of the public point calculated by participant $U_i$; $D$ represents the intermediate value of the second part of the signature, $P$ represents the group public key of the multi-party signature, and $r$ is the first part of the signature, and $s$ is the second part of the signature. The specific process is shown in Fig 1.

### Key generation

The SM2 key generation algorithm is jointly completed by $m$ participants. No one can master the complete signature key, and enter the elliptic curve parameter params. The specific group public key generation process is shown below.

(1) **Parameter initialization of participants (*Setup*)**: $U_i \rightarrow \{k_i, d_i, P_i, K_i\}$

Participant $U_i$ chooses the private key random number $d_i \in [1, n - 1]$, calculates the public key $P_i = d_i[^*]G$, takes any random number $k_i$, and then calculates the intermediate value $K_i = k_i/d_i$ for the public point calculation.

(2) **All participants jointly calculate the group public key (*Key*)**: $\{d_i, G\} \rightarrow \{P(x, y)\}$

The group public key $P$ is calculated jointly using the participant's private key $d_i$ and the generator $G$. All participants are chain-calculated with the group public key point value, where the initial value of $pk_0$ is the generator $G$. When the group public key $P$ is chained, participant $U_1$ encrypts the computed point value data $pk_1$ using the public key of the next participant $U_i$: $pk_1' = Encpk(pk_1, P_i)$, and then digitally signs the encrypted $pk_1$ using $U_1$'s private key $d_1$: $\delta = Sign(pk_1, d_1)$, and finally sends the encrypted data $pk_1'$ and digital signature $\delta$ to the next

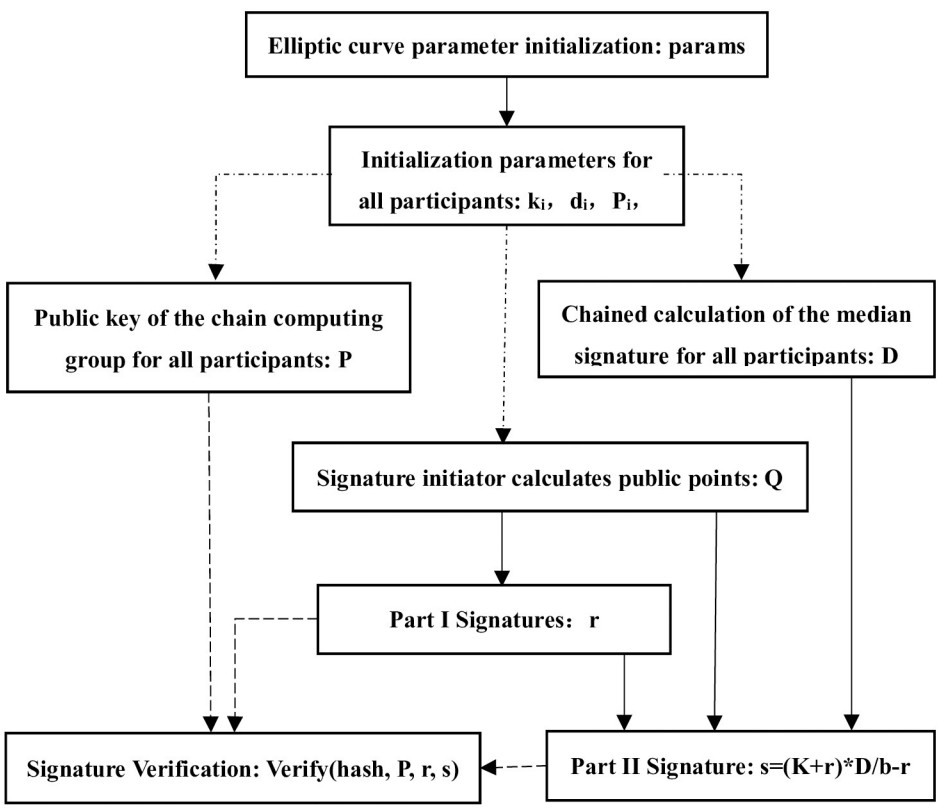

**Fig 1. Multi-party co-signature flow chart.**

participant $U_i$; when the next participant $U_i$ receives the message, decrypts the encrypted data $pk_1{'}$ using the private key $d_i$, and then verifies the digital signature using the encrypted plaintext data, and if the verification passes, proceeds to the next step of computation and sends the encrypted data and digital signature to the next participant. This process can be started from different participants, and the group public key is calculated several times, and the specific operation flow is shown in Fig 2.

The specific steps are as follows:

① Participant $U_1$ initializes the parameters, calculates $pk_1 = d_1^{-1}[*]pk_0$, and then sends the ciphertext data and digital signature of $pk_1$ to participant $U_2$.

② Participant $U_2$ initializes the parameters, first decrypts the received data and verifies the signature. If the verification is passed, calculate $pk_2 = d_2^{-1}[*]pk_1$, and finally send the cip-her-text data and digital signature of $pk_2$ to the participant $U_3$;

．．．．．．

ⓜ Participant $U_m$ initializes parameters, first decrypts the received data and verifies the signature. If the verification is passed, then calculate $pk_m = d_m^{-1}[*]pk_{m-1}$, and the last participant $U_m$ calculates $P = pk_m[*]G$ and publish the group public key, where $P = d_m^{-1}[*]pk_{m-1} - G = \ldots = d_m^{-1} d_{m-1}^{-1} \ldots d_1^{-1}[*]G - G$.

## Co-signature

Any participant can initiate a signature collaboration request as the signature initiator $U_x$, where $x \in [1, m]$, and the public point $Q$ used in the multi-party collaborative signature

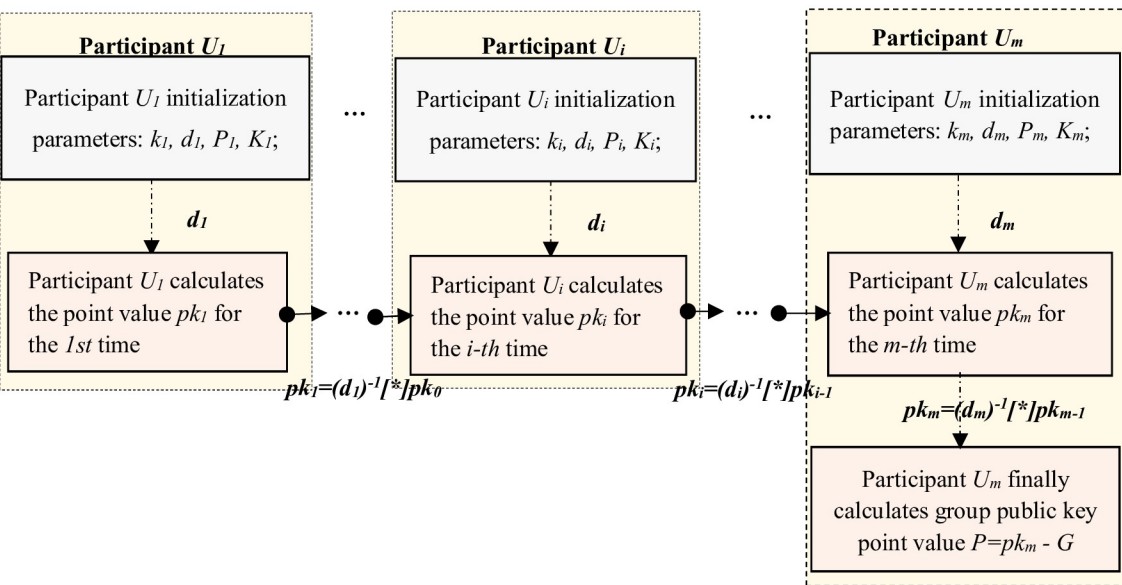

**Fig 2. Key generation flow chart.**

is computed jointly by all participants, and all participants except the signature initiator $U_x$ send the intermediate value $K_i$ of the public point computation to the signature initiator.

(1) **All participants chained calculation of the second part of the signature intermediate value ($Sign_D$):** $\{D_{i-1}, d_i\} \rightarrow \{D_i\}$

The signature initiator $U_x$ selects the private value $b \in [1, n-1]$ and uses the private key $d_x$ to calculate the intermediate transmission value $D_1 = d_x{}^*D_0$; all participants except the signature initiator $U_x$ chain compute the second part of the signature intermediate value using the private key $d_i$, where the initial value of D0 is the private value $b$, which is only known to the signature initiator $U_x$. This process starts with the signature initiator, and there is no sequence among chained computing users. The specific steps are as follows.

① Signature initiator $U_x$ selects the private value $D_0 = b$, calculates $D_1 = d_x{}^*D_0$, and then sends $D_1$ to participant $U_1$.

② Participant $U_1$ uses the private key $d_1$, calculates $D_2 = d_1{}^*D_1$, and then sends $D_2$ to participant $U_2$.

. . . . . .

ⓘ Participant $U_i$ uses the private key $d_i$ to calculate $D_{i+1} = d_i{}^*D_i$, and then sends $D_{i+1}$ to the participant $U_{i+1}$, where $i \in [1, m]$ and $i \neq x$;

ⓜ Participant $U_m$ uses the private key $d_m$ to calculate $D_{m+1} = d_m{}^*D_m$, where $D = D_{m+1} = d_m {}^* D_m = \ldots = d_m d_{m-1} \ldots d_1 {}^* b$, and then the last participant Um returns D to the signature initiator $U_x$.

(2) **Signature initiator $U_x$ calculates the first part of the signature r ($Sign\_r$):** $\{Q, hash\} \rightarrow \{r\}$

① Signature initiator $U_x$ computes $Z_A = H_v(ENTL \| ID_A \| a \| b \| P \| pk)$, where $ID_A$ is the user's distinguishable identifier and $ENTL$ is the length of $ID_A$, and computes $M' = Z_A \| M$;

② Signature initiator $U_x$ computes $e = H_v(M')$ and converts the text $e$ to an integer, where $H_v()$ is a one-way function;

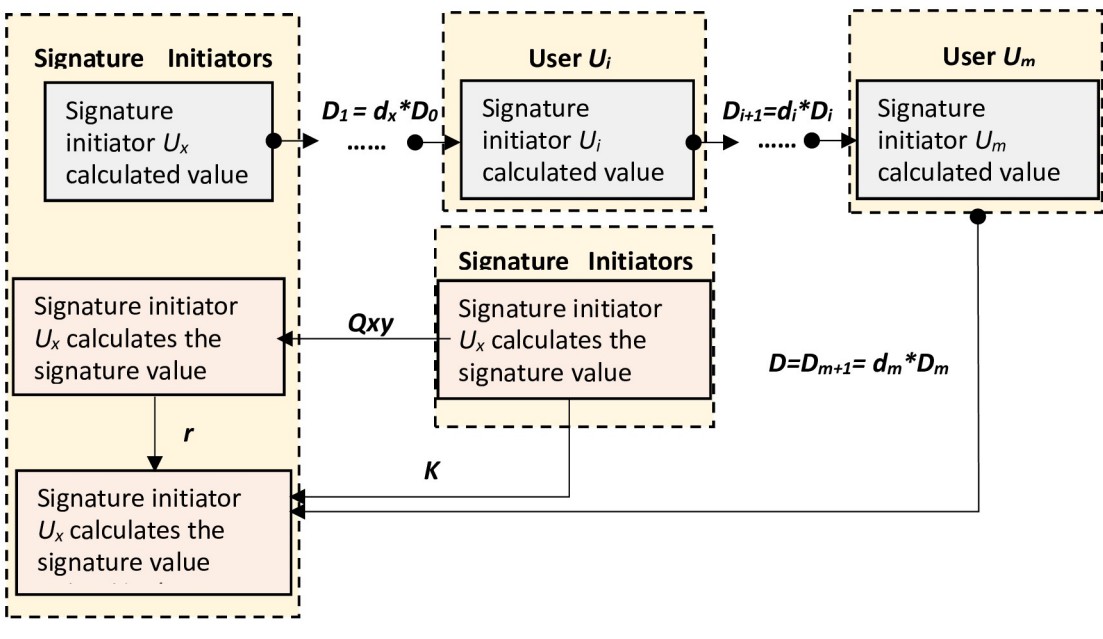

**Fig 3. Multi-signature flow chart.**

③ Signature initiator $U_x$ calculates the value $K = (K_1 + K_2 + \ldots + K_m)$ mod $n$ based on the data $K_i$ sent by each participant, and then calculates the open point $Q = K[^*]G = (x_1, y_1)$ and converts the data type of $x_1$ to an integer;

④ Calculate the signature $r = (e + x_1)$ mod $n$, if $r = 0$ or $r + k = n$, return ③.

(3) **The signature initiator $U_x$ calculates the second part of the signature $s$ (Sign_s):** $\{K, r, D_m, b\} \rightarrow \{s\}$

⑤ After the signature initiator $U_x$ receives the intermediate value $D$ of the second part of the signature, it uses the private value $b$, the self-calculated $K$ value and the first part signature $r$ value to calculate the second part signature $s$ value. Among them, $s = (K + r)^*D/b$. The specific operation process is shown in Fig 3.

(4) **Anyone can verify the signature result ($r'$, $s'$) (Verify):** $\{P, M', r, s\} \rightarrow \{R == r\}$

According to the verification algorithm of SM2 digital signature, only need to calculate the elliptic curve point value $(x'_1, y'_1) = s'[*]G + t[*]P$ and the signature data value $R = e' + x'_1$ mod $n$ in sequence, and finally verify whether $R = r'$ holds, and the specific verification steps of the signature verifier $U_x$ are shown below.

① The verifier separately checks whether $r' \in [1, n-1]$ and $s' \in [1, n-1]$ are established, if not, the verification fails;

② Set $M'' = Z_A \| M'$; compute $e' = H_v(M'')$ and convert the data type of $e'$ to an integer;

③ Convert the data types of $r'$, $s'$ to integers and calculate $t = (r' + s')$ mod $n$. If $t = 0$, the verification fails;

④ Calculate the elliptic curve point $(x'_1, y'_1) = s'[*]G + t[*]P$ and convert the data type of $x'_1$ to an integer;

⑤ Calculate $R = (e' + x'_1)$ mod $n$ and check whether $R = r'$ holds, if it does, the validation passes; otherwise, the validation fails.

## Verification process

Based on the key generation and co-signing process, the correctness of the scheme can be analyzed. The elliptic curve parameters are known: coefficients $a$ and $b$, modulus $p$, generator $G$, and order $n$. In this scheme, each participant $U_i$ has private data private key $d_i$, random number $k_i$, public data $P_i$, and non-private data $K_i$. The specific verification process is shown below.

First, generate the group public key $P$ according to the key generation protocol:

$$P = d_m^{-1}d_{m-1}^{-1}\ldots d_1^{-1}[*]G - G \tag{3}$$

Then, calculate the first part of the signature intermediate value $K$ and the public point $Q$, and the second part of the signature intermediate value $D$:

$$K = (K_1 + \ldots + K_m)\, mod\, n = (\frac{k_1}{d_1} + \ldots + \frac{k_m}{d_m})\, mod\, n \tag{4}$$

$$Q = (x_1, y_1) = K[*]G \tag{5}$$

$$D = d_m d_{m-1}\quad \ldots d_1 * b \tag{6}$$

Finally, the signature $(r, s)$ is generated according to the co-signature protocol rules:

$$r = e + x_1, \ s = \frac{(K+r)*D}{b} - r = (K+r)*d_m\ldots d_1 - r \tag{7}$$

Derive the verification formula $(x_1', y_1') = [s']G + [t]P$ for the SM2 digital signature algorithm.

$$(x_1', y_1') = s'[*]G + t[*]P = (r' + s')d_m^{-1}\ldots d_1^{-1}[*]G - r'[*]G \tag{8}$$

From $s' = (K+r)*D/b - r$, $D = d_m\ldots d_1*b$, we can get:

$$(x_1', y_1') = (r' + \frac{(K+r)*D}{b} - r)d_m^{-1}\ldots d_1^{-1}[*]G - r'[*]G = K[*]G \tag{9}$$

From $(x_1', y_1') = K[*]G$, we know that $x_1'$ is equal to $x_1$, and the final validation calculation value $R = (e + x_1')\, mod\, n$ is exactly the same as the signature value $r = (e + x_1)\, mod\, n$. The validation relation $R = r$ holds, and the signature verification passes. It can be seen that the correctness of the scheme is verified and this scheme is completely feasible.

## Implementation of SM2-based multi-party co-signature scheme

In the implementation of the multi-party co-signing scheme in this paper, the number of specific participants m is determined when a multi-party co-signing scenario is identified in a particular enterprise or department. The implementation of this scheme is divided into 2 main parts: key generation implementation and co-signing implementation. This section will mainly introduce the custom functions and implementation process related to the scheme in this paper, and the specific implementation principles of the scheme are described in Sections 4.2 and 4.3.

## Key generation implementation

The key generation process of SM2 algorithm is mainly done by m participants, which mainly includes the initialization of the participants' parameters and the generation of the public key of the signature group.

**Definition 3. *SM2_User_init*()**: The main function of this function is to allow each user to obtain private data and use it by calling the interface *SM2_User_init*(). In practical applications, both the random number $k_i$ and the private key $d_i$ are generated by a random number generator. This process will be implemented using the function *SM2_User_init*(*Useruser*[]), where the parameters transmitted are the single-user structure *User*, which contains the user identification *uID*, random number $k_i$, private key $d_i$, public key $P_i$, and public point calculation value $K_i$. The input parameter of this function is *null*, the output parameter is the structure *User*, and the return value type is *null*.

**Definition 4. *SM2_Gpubkey_gen*()**: The main function of this function is for each signature participant to chain call the function *SM2_Gpubkey_gen*($uID$, $d_i$, $pk_{i-1}$, $*pk_i$) and generate the point value $pk_m$, where the initial value of the signature group public key $P$ is $-G$, and then call the function *SM2_Point_add*($pk_m$, $*P$) to generate the final signature group public key $P = pk_m - G$. The input parameters of this function are user *uID*, user private key $d_i$, the previous user calculation point $pk_{i-1}$ (the initial value $pk_0$ is the generator $G$), and the output parameter is the user calculation point $pk_i$; the return value type is *int*, and when the return value is 1, it means that the signature group public key is computed successfully, otherwise, the calculation fails.

The implementation process of key generation is shown below.

1. $U_i : SM2\_User\_init(User * U_i) \rightarrow User\ U_i;$

2. $U_1 \rightarrow U_2 : pk_0 = G; pk_1 = SM2\_Gpubkey\_gen(uID_1, d_1, pk_0, *pk_1);$

   ......

   $U_{i-1} \rightarrow U_i : pk_1 = SM2\_Gpubkey\_gen(uID_{i-1}, d_i, pk_{i-1}, *pk_i);$

   ......

   $U_{m-1} \rightarrow U_m : pk_m = SM2\_Gpubkey\_gen(uID_{m=1}, d_m, pk_{m-1}, *pk_m);$

3. $U_m : P = -G; P = SM2\_Point\_add(pk_m, *P);$

The key generation timing chart is shown in Fig 4.

## Co-signature implementation

The SM2 co-signing process will be implemented in four parts in sequence, which mainly including the initialization of the public data, the calculation of the second part of the signature intermediate value *D*, the generation of the first part of the signature *r*, and the generation of the first part of the signature *s*.

**Definition 5**. *SM2_Group_init*(): The main function of this function is to calculate the required public data, which includes the signature calculation value *K* and the public point $Q_{xy}$, where $Q_{xy}$ will be used as the required value for the calculation of the first part of the signature *r* and *K* will be used as the required value for the calculation of the second part of the signature *s*. In this process, each participant calls the value $K_i$ generated by the function *SM2_User_init*(), and sends the calculated value $K_i$ of the public point to the signature initiator for implementation. The input parameters of this function are all the values $K_i$, the output

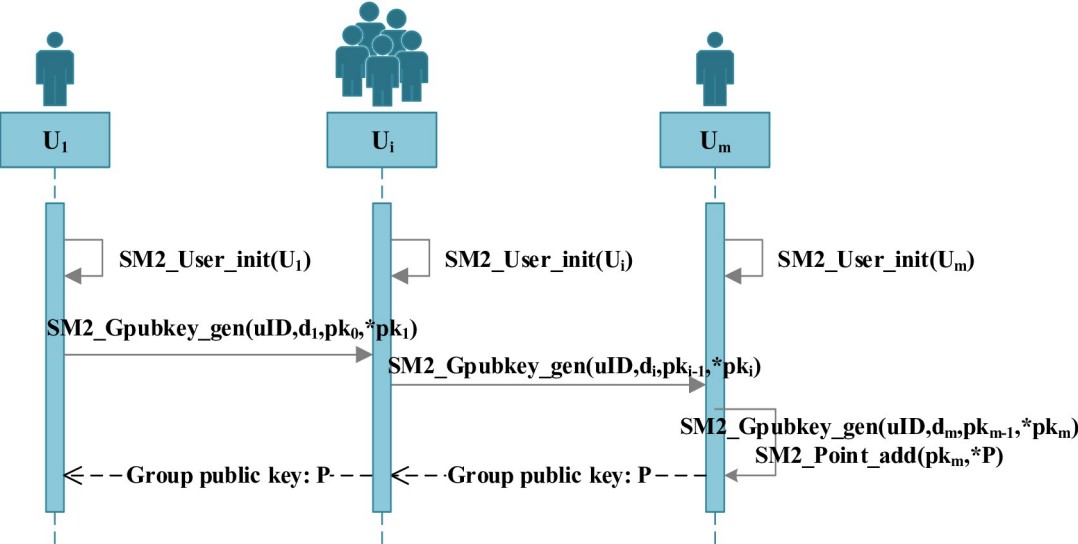

**Fig 4. Key generation timing chart.**

parameters are the signature calculation value $K$ and the public point $Q_{xy}$, the return value type is *int*, when the return value is 1, it means that the signature calculation value and the public point value are successfully calculated, otherwise, The calculation fails.

**Definition 6**. $SM2\_cor\_sign_D()$: The main function of this function is to let each signature participant chain calls the function $SM2\_cor\_sign_D(uID, D_{i-1}, d_i, {}^*D_i)$ and generate the intermediate value $D$ calculated by the private key, and transmits the intermediate value $D$ of the private key calculation obtained by the last participant calling the function $SM2\_cor\_sign\_D()$ to the signature initiator $U_x$. The input parameters of this function are the user *uID*, the previous user calculation point $D_{i-1}$ (the initial value $D_0$ is the calculation data of the signature initiator's optional value $b$ and the private key), the user private key $d_i$, the output parameter is the user calculation point $D_i$, and the return value type is *int*. When the return value is 1, it means that the calculation of the intermediate value D of the private key is successful, otherwise, the calculation fails.

**Definition 7**. $SM2\_cor\_sign\_r()$: The main function of this function is to generate the first part of the signature *r*. This process takes the public point $Q_{xy}$ and the hash value of the message to be signed generated by the signature initiator $U_x$ calling the $SM2\_Group\_init()$ function as input data, and then uses the $SM2\_cor\_sign\_r()$ function to calculate the first part of the signature *r*. The input parameter of this function is the public point $Q_{xy}$, the hash value of the message to be signed, and the output parameter is the first part of the signature value *r*. The return value type is *int*, and when the return value is 1, it means that the first part of the signature is calculated successfully, otherwise, the calculation fails.

**Definition 8**. $SM2\_cor\_sign\_s()$: the main function of this function is to generate the second part of the signature *s*. This procedure will call the $SM2\_cor\_sign\_r()$ function to calculate the value r, the value K generated by the $SM2\_Group\_init()$ function, the calculated value $D$ of the $SM2\_cor\_sign\_D()$ function, and the signature initiator's self-selected value $b$ as input data, and finally the second part of the signature $s$ is calculated using the $SM2\_cor\_sign\_s()$ function. The input parameters of this function are the first part signature *r*, the signature calculation value $K$, the intermediate value $D$ of the private key calculation, and the self-chosen value $b$ of

the signature initiator, and the output parameter is the second part signature value *s*. The return value type is *int*, and when the return value is 1, it means that the second part signature calculation is successful, otherwise, the calculation fails.

The implementation process of the co-signature is shown below, and the message to be signed is *M*.

1. $U_i \rightarrow U_x : K_i = SM2\_User\_init(User * U_i);$

2. $U_x : Q_{xy} = SM2\_Group\_init(K_i, *K, *Q_{xy});$

   $r = SM2\_cor\_sign\_r(hash, Q_{xy}, *r);$

3. $U_x \rightarrow U_1 : D_0 = b; D_1 = SM2\_cor\_sign\_D(uID_x, D_0, d_x, *D_1);$

   $U_1 \rightarrow U_2 : D_2 = SM2\_cor\_sign\_D(uID_1, D_1, d_1, *D_2);$

   ......

   $U_{i-1} \rightarrow U_i : D_i = SM2\_cor\_sign\_D(uID_{i=1}, D_{i-1}, d_{i-1}, *D_i);$

   ......

   $U_m \rightarrow U_x : D = D_{m+1} = SM2\_cor\_sign\_D(uID_m, D_m, d_m, *D_{m+1});$

4. $U_x : s = SM2\_cor\_sign\_s(r, K, D, b, *s);$

The co-signature timing chart is shown in Fig 5.

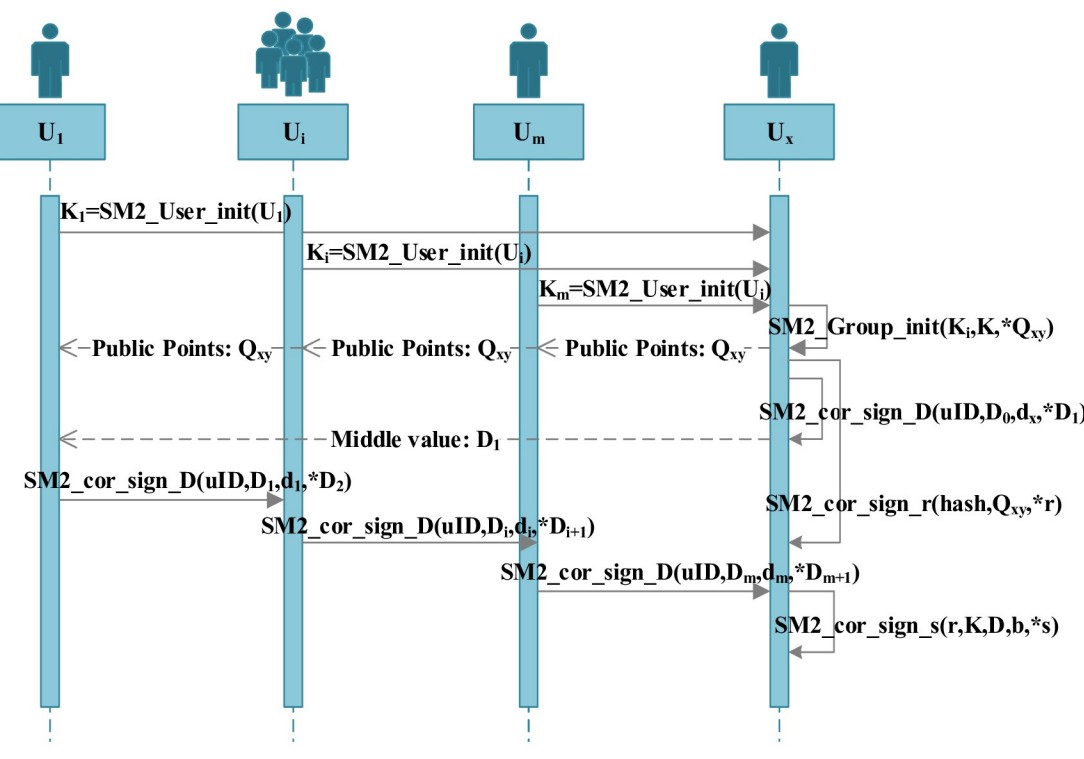

**Fig 5. Co-signature timing chart.**

## Safety analysis

In this section, the provable security based on the original SM2 digital signature scheme is used to analytically prove the security of the multi-party collaborative digital signature scheme designed in this paper.

**Definition 9**. Assuming that the existence of the original SM2 signature scheme $\pi$ is unforgeable under the selective message attack [30], if for any probabilistic polynomial-time adversary in the security model, there exists a negligible function $\mu$ such that for each security parameter $\lambda$, there is $Pr[Sign_{A,\pi}(1^\lambda) = 1] \leq \mu(\lambda)$, that is, the signature scheme $Sign_{A,\pi}$ constructed by adversary after polynomial query, the probability that the signature verification result is 1 is still less than one The negligible function $\mu(\lambda)$, then this scheme is proved to be secure.

The collaborative signature scheme is also known as the distributed signature generation method, and the output digital signature can be effectively verified by the original verification algorithm. Therefore, the existence of the collaborative signature scheme can also be defined as unforgeable.

**Definition 10**. Assume that the existence of the SM2 multi-party co-signature scheme $\Pi$ under the selected message attack is unforgeable, its safety factor is $\kappa$. If for any probabilistic polynomial-time adversary in the security model, adversary passes through the fake signature initiator or corrupting all participants except the signature initiator, there is still a negligible function $\mu$ such that for each security parameter $\kappa$, there is $Pr[DisSign_{A,\Pi}(1^\kappa) = 1] \leq \mu(\kappa)$, the signature scheme $DisSign_{A,\Pi}$ constructed with joint participation of adversary has a probability of signature verification result of 1 less than a negligible function $\mu(\kappa)$, then the multi-party collaborative signature scheme is proved to be secure.

## Proof of corruption of signature initiator

In the whole SM2 multi-party collaborative (at least three-party) digital signature process, adversary corrupts the server of one participant who is a legitimate member of the multi-party collaboration and controls this server to initiate the collaborative signature application and obtains multi-party data only during the participation in the collaborative signature process, at which time the multi-party collaborative scheme is similar to the two-party collaborative signature scheme, but the data obtained is the product of the private keys of multiple participants. Adversary acts as the signature initiator $U_A$, and adversary tries to obtain the other party's private information through simulator $\zeta$ to get hold of the corrupted party's key to forge the signature attack on the original scheme. The specific simulation scheme is shown below.

**(1) Simulation key generation.** ① The adversary $U_A$ initiates the calculation of the group public key $P$. The adversary $U_A$ calculates the point value $pk_1$ several times for encryption and digital signature processing, and sends the encrypted data $pk_1'$ and digital signature $\delta_1$ to the next participant $U_i$.

② When the next participant $U_i$ receives the ciphertext and digital signature, it verifies the ciphertext by digital signature and recalculates the data until the last user calculates and discloses the value of group public key $P$.

Therefore, even if the adversary $U_A$ uses a different value point $pk_1'$ to participate in the calculation of the group public key point value $P'$, the two points $pk_1'$ and $P'$ on the elliptic curve are known, guaranteed by the elliptic curve discrete logarithm difficulty problem, this process still cannot obtain any private information about the private key $d_x$ in polynomial time.

**(2) Simulation of co-signature.** ① The adversary $U_A$ initiates a co-signature request. The adversary $U_A$ chooses any private value $b' \in [1, n-1]$, chooses to use the private key random number $d_x'$ to calculate the intermediate transmission value $D_1 = d_x' * D_0$, and sends $D_1$ to the

next user for chain calculation, and the last participant calculates the intermediate value $D_m = d_m d_{m-1} \ldots d_x' \ldots d_1 * b'$ and returns it to the adversary $U_A$, who finally obtains the product of other users' private keys by the private value $b'$ and the private key random number $d_x'$ as: $d_x' = d_m d_{m-1} \ldots d_{x+1} d_{x-1} \ldots d_1$.

② The adversary $U_A$ calculates the value of $K$ and the value of the public point $Q$. The co-signing participants $U_i$ send their respective calculated intermediate values $K_i$ to the adversary $U_A$, and the adversary $U_A$ receives $K_i$ also cannot calculate its specific factorized random number $k_i$ value and private key $d_i$ value, and the adversary $U_A$ finally obtains the valid data as the intermediate values of other participants: $K_1, \ldots, K_{x-1}, K_{x+1}, \ldots K_m$, and the sum of the median values of other participants $K_x' = K_1 + \ldots + K_{x-1} + K_{x+1} + \ldots + K_m$. The calculated value $K$ and the public point $Q$ value are ultimately determined by the value of the private key $d_x'$;

③ If the adversary $U_A$ uses the private key random number $d_x'$, product value $D_x'$, sum value $K_x'$ and point value $Q_x'$ to calculate the value $K' = (K_x' + K_x)$ and public point value $Q' = K'[*] G$, finally calculate the intermediate value $D' = d_x' * d_x$ and the signature value $r' = e + Q' -> x_1', s = (K' + r')^* D'/b' - r'$.

④ Finally, the previous $r'$ and $s'$ are verified by any user.

$$
\begin{aligned}
(x_1'', y_1'') \quad &= R' = s'[*]G + t[*]P = s'[*]G + (r' + s')[*]P \\
&= s'[*]G + (r' + s')(d_m^{-1} \ldots d_1^{-1}[*]G - G) \\
&= (r' + \frac{(K' + r') * D}{b'} - r')d_m^{-1} \ldots d_1^{-1}[*]G - r'[*]G \\
&= (K_x' + K_x) * d_x' * d_x^{-1}[*]G + r' * d_x' * d_x^{-1}[*]G - r'[*]G
\end{aligned}
$$
(10)

From the verification results and simulated key generation, it is clear that the probability that the private key random number $d_x'$ is equal to the original private key value $d_x$ is negligible, and the adversary $U_A$ uses the private key random number $d_x'$ to sign, and the signature verification result fails, that is, $Pr[DisSign_{A,\Pi}(1^\kappa) = 1] \leq \mu(\kappa)$ holds constantly. Therefore adversary $U_A$ corrupts a participant's server counterfeit signature initiator to forge a signature fails.

## Proof of maximum corruption

In the entire SM2 multi-party collaborative digital signature process, adversary corrupts the servers of as many collaborative legitimate members as possible (up to $m$-1 participants), at which point the multi-party collaborative signature scheme is equivalent to a two-party collaborative signature containing signature initiator $U_1$ and collaborative participant $U_A$, while adversary corrupts collaborative participant $U_A$, which can be up to $m$-1 participants at this point, and adversary $U_A$ attempts to multiple signature tests to obtain the other party's information and forge the signature. The specific simulation scheme is shown below.

**(1) Simulated key generation.** ① The signature initiator $U_1$ initiates the calculation of the group public key $P$. $U_1$ encrypts and digitally signs the point value $pk_1 = d_1^{-1}[*]G$ calculated by itself and sends the encrypted data of $pk_1$ and the digital signature $\delta_1$ to the co-participant $U_A$.

② When the corrupted participant $U_A$ receives the ciphertext and digital signature, $U_A$ only knows the generated meta point value $G$ and received point value $pk_1$ or the original group public key point value $P$ and received point value $pk_1$, and it is known that the two points on

the elliptic curve cannot obtain the valid privacy information; $U_A$ continues the calculation by the private key random number $d_2'$ and finally gets the group public key $P' = d_2'^{-1} d_1^{-1} [*] G$.

Therefore, the adversary UA cannot obtain any valid information during the simulation key generation process.

**(2) Simulated co-signature.** ① The signature initiator $U_1$ initiates a co-signature request. The signature initiator $U_1$ can choose the private value $b \in [1, n-1]$, and use the private key $d_1$ to calculate the intermediate transmission value $D_1 = d_1 * b$, and send $D_1$ to the corrupt cooperative user $U_A$ for chain calculation.

② The corrupt party $U_A$ only participates in the calculation of the intermediate value $D$ once during the signature process, and only obtains the transmission data $D_1$ of the signature initiator, and calculates the signature $r' = e + Q' - >x_1'$, $s = (K' + r')^* D'/b' - r'$; the point value $Q'$ is the public data, $K'$ and $b'$ are the data held by $U_1$, so it is impossible to forge the signature $s$.

From the simulated co-signature process, it is clear that the adversary $U_A$ cannot obtain valid information when impersonating a co-signature participant. Therefore, adversary fails to obtain information to forge signatures by corrupting the servers of multiple participants.

| **Algorithm 1**: SM2_sign0 | **Algorithm 2**: SM2_signN |
|---|---|
| **Input**: *hash*, *rand*, *prikey* | **Input**: *hash*, *K*, *D*, *b* |
| **Output**: *sign* | **Output**: *sign* |
| int SM2_sign0(hash, rand, prikey, *sign){ | int ec_SM2_signN(hash, K, D, b, *sign){ |
| 1. int res = 0; | 1. int res = 0; |
| 2. mpz_mod(rand, rand, Gn); | 2. point_mult_naf(&group, &base, K, &Q); |
| 3. if (mpz_sgn(rand) == 0) | 3. mpz_mod(gmp_r, Q.X, group.Gn); |
| 4.    mpz_set_ui(rand, 1); | 4. if (mpz_sgn(gmp_r) == 0) |
| 5. point_mult_naf(&group, &base, rand, &Q); | 5.    goto end; |
| 6. mpz_mod(tmp_r, Q.X, group.Gn); | 6. mpz_add(r, gmp_r, hash); |
| 7. if (mpz_sgn(tmp_r) == 0) | 7. mpz_mod(sign->X, r, group.Gn); |
| 8.    goto end; | 8. mpz_add(tmpK, K, r); |
| 9. mpz_add(r, tmp_r, hash); | 9. mpz_mod(gmp_D, tmpK, group.Gn); |
| 10. mpz_mod(sign->X, r, group.Gn); | 10. if (mpz_sgn(gmp_D) == 0) |
| 11. mpz_add_ui(tmp1, prikey, (long)1); | 11.    goto end; |
| 12. mpz_invert(invert, tmp1, group.Gn); | 12. mpz_mul(gmp_S, gmp_D, D); |
| 13. mpz_mul(tmp2, sign->X, prikey); | 13. mpz_mod(S, gmp_S, group.Gn); |
| 14. mpz_mod(tmp3,tmp2, group.Gn); | 14. mpz_invert(invert, b, group.Gn); |
| 15. mpz_sub(tmp4, rand, tmp3); | 15. mpz_mul(gmp_s, S, invert); |
| 16. mpz_mod(tmp_sub, tmp4, group.Gn); | 16. mpz_mod(gmp_s, gmp_s, group.Gn); |
| 17. if (mpz_sgn(tmp_sub) == 0) | 17. mpz_sub(s, gmp_s, gmp_r); |
| 18.    goto end; | 18. mpz_mod(sign->Y, s, group.Gn); |
| 19. mpz_mul(tmp, invert, tmp_sub); | 19. if (mpz_sgn(tmps) == 0) |
| 20. mpz_mod(sign->Y, tmp, group.Gn); | 20.    goto end; |
| 21. res = 1; | 21. res = 1; |
| 22. end: | 22. end: |
| 23. return (res); } | 23. return (res); } |

## Experimental results and performance analysis

In this section, we analyze and evaluate the signature and verification results of the multi-party collaborative signature scheme with different number of participants for the multi-party collaborative signature scheme proposed in Section 4.3, and compare the implementation and verification runtime of the original SM2 scheme with different multi-party collaborative signatures. The implementation of the original SM2 scheme and the multi-party collaborative digital signature algorithm are shown in Algorithm 1 and Algorithm 2, respectively.

### Experimental environment

In this paper, we construct a model based on the multi-party co-signing scheme of SM2 algorithm. The experimental PC operating system is win10 operating system, the processor is Intel (R) Core(TM) i7-5500U CPU @ 2.40GHz 2.39GHz, 8G RAM; the large integer library gmp-6.2.0 is selected for implementation; the main programming language is C; the platform used to implement the algorithm is Visual Studio 2019. In addition, the system parameters are the elliptic curve parameters recommended by the State Secret.

### Experimental results

**Anti-forgery attacks.** Multi-party co-signing is inherently resistant to forgery attacks, During the signature process, the complete private key does not appear in the memory of any party, eliminating the risk of forgery of digital signatures due to private key leakage, and the complete signature cannot be generated without the participation of any party after the public key of the co-signing group is determined. In addition, the identity of each user is legitimate at the time of signature group public key generation, even if the forger forges the signature by impersonating a node in the signature process, the forged signature cannot pass the verification in the final signature verification process with the participation of the group public key, so the forger fails to forge the signature. Taking 3-party collaboration as an example, assume that the authenticated collaborative participants are userA$\{d_A, P_A\}$, userB$\{d_B, P_B\}$, userC$\{d_C, P_C\}$, and the generated legitimate group public key is $P = d_A^{-1}d_B^{-1}d_C^{-1}[*]G - G$. The counterfeiter forgerA attacks the participant userA, and forges information such as the same identity $ID_A$, the same public key $P_A$ (the private key is different $d_X$) and other information. The steps for forging the signature are as follows:

**(1) Forger forgerA requests participant chain to calculate the second part of the signature intermediate value ($Sign\_D$)**

① ForgerA initiates a signature, first selects the private value $D_0 = b$, calculates $D_1 = d_x{}^*D_0$, and then sends $D1$ to the participant userB;

② The participant userB uses the private key dB to calculate $D_2 = d_B{}^*D_1$, and then sends $D_2$ to the participant userC;

③ The participant userC uses the private key $d_C$ to calculate $D_3 = d_C{}^*D_2$, where $D = D_3 = d_C{}^*d_B{}^*d_X{}^*b$, and finally returns $D$ to the signature initiator forgerA.

**(2) The forger forgerA calculates the first part of the signature $r$ ($Sign\_r$)**

① ForgerA calculates $Z_A = H_v(ENTL\|ID_A\|a\|b\|P\|pk)$, and calculates $M' = Z_{A\|M}$;

② The counterfeiter forgerA calculates $e = H_v(M')$, and converts the text $e$ to an integer, where $H_v()$ is a one-way function;

③ The forger forgerA calculates the value $K = (K_X + K_A + K_B)modn$ based on the data $K_i$ sent by each participant, where $K_X = k_X/d_X$, and then calculates the public point $Q = K[*]G = (x_1, y_1)$, and convert the data type of $x_1$ to integer.

④ Calculate the signature r = (e+x1) mod n, and return ③ if r = 0 or r+k = n.

**(3) Forger forgerA calculates the second part of signature $s$ ($Sign\_s$)**

⑤ ForgerA uses the signature intermediate value $D$, private value $b$, self-calculated $K$ value and the first part of the signature $r$ value to calculate the second part of the signature s value, where $s = (K + r)^* D/b - r = (K + r)^* d_C\, d_B\, d_X - r$.

**(4) Verification of the forged signature result ($r'$, $s'$)**

Verify the forged digital signature by deriving the verification formula $(x_1', y_1') = [s']G + [t]P$, where $P = d_A^{-1}d_B^{-1}d_C^{-1}[*]G - G$, $t = (r' + s') mod\, n$. The detailed verification steps of the signature are described in Section 4.4.

$$
\begin{aligned}
Q' = (x_1', y_1') &= s'[*]G + t[*]P = s'[*]G + (r' + s')[*]P \\
&= s'[*]G + (r'+s')(d_A^{-1}d_B^{-1}d_C^{-1}[*]G - G) = (r'+s')d_A^{-1}d_B^{-1}d_C^{-1}[*]G - r'[*]G \\
&= (r' + (K + r') * d_C d_B d_X - r')d_A^{-1}d_B^{-1}d_C^{-1}[*]G - r'[*]G \qquad (11) \\
&= (K * d_X/d_A + r' * d_X/d_A)[*]G - r'[*]G \\
&= (K * d_X/d_A)[*]G + (r' * d_X/d_A - r')[*]G
\end{aligned}
$$

It can be seen from the verification result of the forged signature that $Q' \neq Q$, $x_1' \neq x_1$, then $R \neq r'$, that is, the verification fails and the forged signature fails.

**Feasibility and flexibility validation.**   In order to verify the feasibility and flexibility of this solution, we choose a specific application scenario: an enterprise produces an important product for a large number of users, and needs qualified manufacturers of semi-finished products, and qualified semi-finished products need qualified manufacturers of components, qualified parts manufacturers need qualified raw material suppliers to provide, which forms a supply chain. In order to ensure that the quality of the final product is recognized and accepted by the user, the company needs the company in the supply chain to collaborate to sign a product quality commitment to users, and the user can verify the signature to trust the quality of the product. Because once the user's verification is passed, it means that the supply chain of the product is trustworthy.

In response to this scenario, we separately set the private key and random number of the SM2 digital signature, and the private keys and random numbers of multiple companies in the industry chain. In order to facilitate the experiment, we take the same values for the private keys and random numbers of multiple enterprises in the industry chain as the original SM2 digital signature scheme: $d_A$ = "0x128B2FA8 BD433C6C 068C8D80 3DFF7979 2A519A55

**Fig 6. Original SM2 digital signature and verification data.**

```
N = 3

Message:    message digest

Hash:      c522a942 e89bd80d 97dd666e 7a5531b3 6188c981 7149e9b2 58dfe51e ce98ed77

P.X        870ef0dd 59423096 b10e5199 7455a9cd 8c3a4cd8 28e9b67e d673f053 43a94351
P.Y        02ec71b9 430dbea3 d3f23f64 12612afd 100157e9 f681b47d 17e58b3e dc17140d

r:         48a6a3aa a11cbfe2 725c86a8 4930d759 c3ebd934 825c2469 56770e40 023237fb

s:         0f4ad3dc b256d593 0f2f8832 f155dfd8 3d6caf86 903fe7e9 b6f65d89 d9bdda4d

*******Verify the sign Success!********
```

**Fig 7. 3-party co-digital signature and verification data.**

171B1B65 0C23661D 15897263"; $k$ = "0x6CB28D99 385C175C 94F94E93 4817663F C176D925 DD72B727 260DBAAE 1FB2F96F".

(1) The plaintext message digest of the original SM2 digital signature scheme uses the SM3 algorithm to calculate the hash $e$(Hash) and the public key $P(X, Y)$, and the digital signature ($r$, $s$) and signature verification results are shown in Fig 6.

(2) The SM3 algorithm is used to calculate the hash $e$(Hash) and group public key $P(X, Y)$, as well as the digital signature ($r$, $s$) and signature verification results for the plaintext "message digest" of the SM2 industry chain multi-enterprise collaborative digital signature scheme. The results of co-signatures are shown in Figs 7–9 for 3, 8, and 23 parties.

From the above experimental results, it can be seen that this scheme has strong flexibility, and the participants of the supply chain can be any $n$ parties such as 3 parties or 8 parties to participate in the collaborative signature, which can meet the requirements of any multi-party hierarchical evaluation of the same document in electronic transactions, making the multi-party collaborative signature scheme have stronger applicability. In addition, this scheme can obtain legal signature results for different number of supply chain participants and the verification result is correct, which shows the correctness and feasibility of this scheme. In addition,

```
N = 8

Message:    message digest

Hash:      c522a942 e89bd80d 97dd666e 7a5531b3 6188c981 7149e9b2 58dfe51e ce98ed77

P.X        6866ec37 14c63880 2f387bdb df049b4a e7c8db5f 034ac78c f6f3b929 a327a84b
P.Y        5ec54fe4 d040e5f9 da00a320 ed3a2d50 ffad0fe7 09fa49ce 71f12463 2a146800

r:         671f454e e0955a6f ccfc41af 18a783f3 af23065c 2d7739f0 5e5fadc8 8abd0e35

s:         09a5557f 7795a6aa c88396d8 32c382c3 7e859b0a cce618d3 db163286 9f39f660

*******Verify the sign Success!********
```

**Fig 8. 8-party co-digital signature and verification data.**

**Fig 9. 23-party co-digital signature and verification data.**

for the $n$-party co-signing scheme, unless the hacker obtains the signature key of the $n − 1$ signer, the legitimate signature cannot be forged, so the multi-party co-signing scheme in this paper also has sufficient security.

**Performance analysis.**　For the differences caused by running in different time periods in the native environment, firstly, the original SM2 scheme will be carried out several times with multiple co-signers at the same time, and then the running time of multiple signatures of the original SM2 scheme (the difference in the values taken should not be too large) will be taken, and finally the average value will be used as the standard value for floating values, and the running time of multiple co-signers in the corresponding range of floating values will be selected. For different parties, 100 tests were conducted, and the number of multiparty co-signers was taken as 3, 8, 23, etc., and then the average value of the running time was calculated. The average running time of the digital signature interface and signature verification interface of the original SM2 scheme and the multi-party collaboration algorithm are shown in Fig 10.

The above data counts the average running time of the digital signature interface of the original SM2 scheme and the multi-party collaboration algorithm (that is, Algorithm 1 and Algorithm 2), and after running Algorithm 1 and Algorithm 2 several times simultaneously, changing only the number of supply chain participants in the collaborative signature each time, and finally taking the running time of the original SM2 signature as the standard value to take the running time of multiple interfaces and calculate the average value. From the above statistical results, it can be seen that there is no additional burden on the performance of the signature scheme when increasing the number of multi-signers, because the signature time of Algorithm 2 here is only the interface time for the final calculation of the signature result by the tested signature initiator, and the number of supply chain participants does not correlate with the signature time, so the signature times of Algorithm 1 and Algorithm 2 are almost the same. In addition, for the results of multi-party co-signing, signature verification has been performed using the standard SM2 verification interface, and the verification in Section 7.2.1 experimental results passed proving that multi-party co-signing is feasible, flexible and universally applicable to application scenarios such as multi-user and multi-device digital signatures. Thus, although there is a certain amount of computing time and data transmission time when calculating the intermediate values before the signature of this scheme, this scheme is still highly feasible and has wide application prospects.

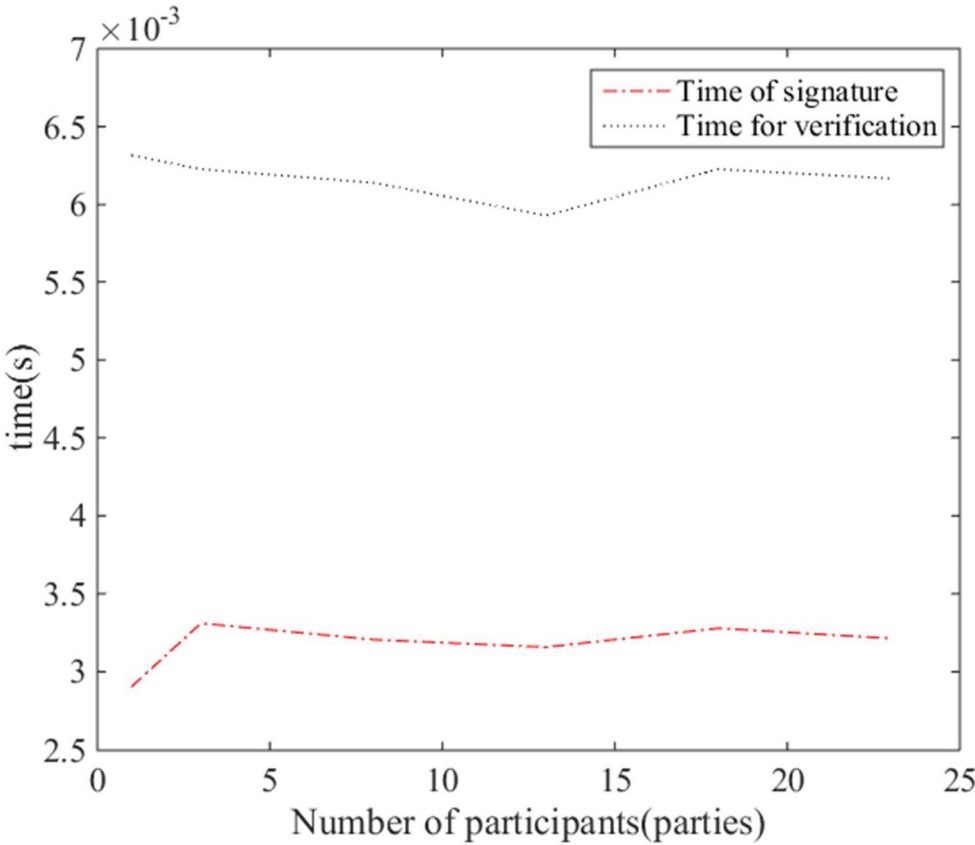

**Fig 10. Average runtime of SM2 scheme and multi-party co-digital signature and signature verification.**

## Conclusion

In order to prevent the leakage of the user's key stored in the client in the mobile Internet environment and to meet the needs of multiple entities operating on the same file, this paper designs a multi-party collaborative signature scheme based on SM2 based on the idea of collaborative signature. It allows users to hold their own private key values and jointly complete the SM2 digital signature without revealing the key, and both the group public key and signature must be jointly generated by multiple parties, and the complete signature key cannot be recovered in the process, fully guaranteeing the security of the signature key. In addition, we attribute the security of co-signatures to standard SM2 signatures, thus proving the security of our proposed protocol, that is, if the original SM2 is probabilistic polynomial-time unforgeable, then our protocol also satisfies probabilistic polynomial-time unforgeability. Finally, this article has completed the test and evaluation for the correctness and performance of the scheme. The experimental results show that the multiparty co-signing takes slightly more time than the original SM2 signature once for the same device in an approximately consistent environment, but the signature verification interface and verification time are consistent. The co-signature does not change the format of the signature result and the signature verification algorithm. It has high compatibility with the standard SM2 digital signature algorithm, so it has a wide application prospect in practical applications, especially in the multi-user and multi-device application scenario with strong scalability and practicality.

## Author Contributions

**Conceptualization:** Liang Tan, Zhongzhu Liu.

**Data curation:** Xinglin Shang, Liping Zou, Zhongzhu Liu.

**Formal analysis:** Liang Tan.

**Funding acquisition:** Liang Tan.

**Investigation:** Liang Tan.

**Methodology:** Liang Tan.

**Project administration:** Liang Tan.

**Resources:** Zhongzhu Liu.

**Software:** Hekun Yang, Yi Wen.

**Validation:** Zhongzhu Liu.

**Visualization:** Zhongzhu Liu.

**Writing – original draft:** Liang Tan, Zhongzhu Liu.

**Writing – review & editing:** Liang Tan, Zhongzhu Liu.

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
