## [Decision Letter · Decision Letter 0]

2 Feb 2022

PONE-D-21-40703Multi-party co-signature scheme based on SM2PLOS ONE

Dear Dr. zhongzhu,

Thank you for submitting your manuscript to PLOS ONE. After careful consideration, we feel that it has merit but does not fully meet PLOS ONE’s publication criteria as it currently stands. Therefore, we invite you to submit a revised version of the manuscript that addresses the points raised during the review process. Please submit your revised manuscript by Mar 19 2022 11:59PM. If you will need more time than this to complete your revisions, please reply to this message or contact the journal office at plosone@plos.org. Please include the following items when submitting your revised manuscript:A rebuttal letter that responds to each point raised by the academic editor and reviewer(s). You should upload this letter as a separate file labeled 'Response to Reviewers'.A marked-up copy of your manuscript that highlights changes made to the original version. You should upload this as a separate file labeled 'Revised Manuscript with Track Changes'.An unmarked version of your revised paper without tracked changes. You should upload this as a separate file labeled 'Manuscript'.If applicable, we recommend that you deposit your laboratory protocols in protocols.io to enhance the reproducibility of your results. Protocols.io assigns your protocol its own identifier (DOI) so that it can be cited independently in the future. For instructions see: https://journals.plos.org/plosone/s/submission-guidelines#loc-laboratory-protocols. Additionally, PLOS ONE offers an option for publishing peer-reviewed Lab Protocol articles, which describe protocols hosted on protocols.io. Read more information on sharing protocols at https://plos.org/protocols?utm_medium=editorial-email&utm_source=authorletters&utm_campaign=protocols.

We look forward to receiving your revised manuscript.

Kind regards,

Pandi Vijayakumar, Ph.D

Academic Editor

PLOS ONE

Journal Requirements:

"Unfunded studies"

"NO authors have competing interests"

Reviewers' comments:

Reviewer's Responses to Questions

**Comments to the Author**

1. Is the manuscript technically sound, and do the data support the conclusions?

Reviewer #1: Partly

Reviewer #2: Yes

Reviewer #3: Yes

2. Has the statistical analysis been performed appropriately and rigorously? 

Reviewer #1: Yes

Reviewer #2: Yes

Reviewer #3: Yes

3. Have the authors made all data underlying the findings in their manuscript fully available?

Reviewer #1: Yes

Reviewer #2: Yes

Reviewer #3: Yes

4. Is the manuscript presented in an intelligible fashion and written in standard English?

Reviewer #1: Yes

Reviewer #2: Yes

Reviewer #3: Yes

5. Review Comments to the Author

Reviewer #1: Only 10% of reference in this manuscript is recent work others are too old.

Authors has to increase recent references in to 20%.

Align the paper to journal format.

No keyword or indexed terms in your research work.

Reviewer #2: This article proposed cryptography mechanism for multiple user signing. This will ensure the product verification in many applications. Kindly answer the below questions.

1. The scheme proposed here matches with blockchain technology, except mutiuser sign. But, aim of the work can be accomplished by blockchain permissionless (private) scheme, in which only the authenticated users can verify. Pls clarify.

2. The performance analysis indicate only the runing time of the SM2 scheme. There are many security metrics available.

Reviewer #3: This work proposes a multi-party collaborative signing technique based on the SM2 digital signature algorithm from the GM/T003-2012 standard "SM2 Elliptic Curve Public Key Cryptography." This approach entails several (more than 2) participants working together in an interactive manner to generate the signature group public key and valid signature, while ensuring that no user knows the signature key other than their own during the signing process. We use the GMP library to implement this strategy. The experimental findings show that this approach is not only more flexible, but also more secure and trustworthy in the multi-device collaborative signing application scenario. Furthermore, the time it takes for numerous participants to construct signatures in this method is similar to that of the original SM2 signature, and the time it takes for signature verification is similar to that of the original SM2 signature. Authors should, however, handle the following concerns.

• In this survey, the authors missed the case study. I suggested authors to add case study about signature scheme so that the international readers could follow the work and understand easily.

1. Privacy-Protection Scheme Based on Sanitizable Signature for Smart Mobile Medical Scenarios

2. Efficient NTRU lattice-based certificateless signature scheme for medical cyber-physical systems

3. A practical group blind signature scheme for privacy protection in smart grid

4. Efficient and provably secure multireceiver signcryption scheme for multicast communication in edge computing

5. TC-PSLAP: Temporal Credential-Based Provably Secure and Lightweight Authentication Protocol for IoT-Enabled Drone Environments

• I suggested authors should check the typo errors.

After answering above queries, the review paper may be considered for publication. I can recommend for Minor revision.

6. PLOS authors have the option to publish the peer review history of their article (what does this mean?). If published, this will include your full peer review and any attached files.

Reviewer #1: **Yes: **Dr. D. RAJESH

Reviewer #2: **Yes: **Saravanan K

Reviewer #3: No

---

## [Author Response · Author response to Decision Letter 0]

18 Apr 2022

We thank the editor and the reviewers for their insightful comments. We are glad that they have found the current research to be valuable, the method used reliable to provide novel results and the discussion substantial to explain the results. In sum, we thank the reviewers for noting that we have filled the research gap by asking an important question that has not yet been thoroughly addressed in the existing literature.

We have further revised the manuscript in light of the review comments. Below, we summarise the major changes we have made to the manuscript, and then we detail our responses to the review comments point by point.

We numbered the comments made by reviewers and started with bold letters " Q1" for the ﬁrst reviewer, "Q2" for the second reviewer, "Q3" for the third reviewer, and "Q4" for the fourth reviewer. In addition, responses to the comments are started with the bold letter "Response to *". In order to make the response letter and revised manuscript easier to read for the editor and reviewers, the changes are marked in "red color" for reviewers' comments. Finally, we change the order of authors.

Data Availability Statement:

All relevant data in this study are available at https://doi.org/10.5064/F6ULCCQA

Revisions

Thanks to the reviewers for the questions raised during the review process, we have made the following modifications in response to the additional requirements for revision comments.

1. The style of the manuscript was revised, as well as the naming of the documents and other issues.

2. Reviewed the citation format of references and revised the format of some of the literature; in addition, old literature was replaced with new ones and relevant literature was added.

3. Added a section "Acknowledgments" to the manuscript, in which the source of funding for the study is described.

4. Regarding the issue of "Competing Interests", there are no competing interests among the authors in this study.

5. Submit the corresponding minimal data set of the results described in the manuscript.

6. In the manuscript, a detailed description of the review comments and the corresponding changes were added, as described in "Responses to Reviewer".

---

## [Decision Letter · Decision Letter 1]

26 Apr 2022

Multi-party co-signature scheme based on SM2

PONE-D-21-40703R1

Dear Dr. Liu,

We’re pleased to inform you that your manuscript has been judged scientifically suitable for publication and will be formally accepted for publication once it meets all outstanding technical requirements.

Kind regards,

Pandi Vijayakumar, Ph.D

Academic Editor

PLOS ONE

Additional Editor Comments (optional):

Reviewers' comments:

Reviewer's Responses to Questions

**Comments to the Author**

1. If the authors have adequately addressed your comments raised in a previous round of review and you feel that this manuscript is now acceptable for publication, you may indicate that here to bypass the “Comments to the Author” section, enter your conflict of interest statement in the “Confidential to Editor” section, and submit your "Accept" recommendation.

Reviewer #1: All comments have been addressed

Reviewer #2: All comments have been addressed

2. Is the manuscript technically sound, and do the data support the conclusions?

Reviewer #1: Yes

Reviewer #2: Yes

3. Has the statistical analysis been performed appropriately and rigorously? 

Reviewer #1: Yes

Reviewer #2: Yes

4. Have the authors made all data underlying the findings in their manuscript fully available?

Reviewer #1: Yes

Reviewer #2: Yes

5. Is the manuscript presented in an intelligible fashion and written in standard English?

Reviewer #1: Yes

Reviewer #2: No

6. Review Comments to the Author

Reviewer #1: Manuscript is organized and corrected as per the comments. so it may be Accepted.

Reviewer #2: author addressed the queries and revised the paper. the revised paper has the proper strcture and narration

7. PLOS authors have the option to publish the peer review history of their article (what does this mean?). If published, this will include your full peer review and any attached files.

Reviewer #1: **Yes: **Dr. D. Rajesh

Reviewer #2: **Yes: **Saravanan K

---

## [Editor Report · Acceptance letter]

20 Jun 2022

PONE-D-21-40703R1 

Multi-party co-signature scheme based on SM2  

Dear Dr. Liu:

I'm pleased to inform you that your manuscript has been deemed suitable for publication in PLOS ONE. Congratulations! Your manuscript is now with our production department. 

Kind regards, 

on behalf of

Dr. Pandi Vijayakumar 

Academic Editor

PLOS ONE